# Structural Characteristics of Heparin Binding to SARS-CoV-2 Spike Protein RBD of Omicron Sub-Lineages BA.2.12.1, BA.4 and BA.5

**DOI:** 10.3390/v14122696

**Published:** 2022-12-01

**Authors:** Deling Shi, Changkai Bu, Peng He, Yuefan Song, Jonathan S. Dordick, Robert J. Linhardt, Lianli Chi, Fuming Zhang

**Affiliations:** 1National Glycoengineering Research Center, Shandong University, Qingdao 266237, China; 2Center for Biotechnology and Interdisciplinary Studies, Rensselaer Polytechnic Institute, Troy, NY 12180, USA

**Keywords:** SARS-CoV-2, Omicron, spike protein RBD, heparin, pentosan polysulfate, mucopolysaccharide polysulfate

## Abstract

The now prevalent Omicron variant and its subvariants/sub-lineages have led to a significant increase in COVID-19 cases and raised serious concerns about increased risk of infectivity, immune evasion, and reinfection. Heparan sulfate (HS), located on the surface of host cells, plays an important role as a co-receptor for virus–host cell interaction. The ability of heparin and HS to compete for binding of the SARS-CoV-2 spike (S) protein to cell surface HS illustrates the therapeutic potential of agents targeting protein–glycan interactions. In the current study, phylogenetic tree of variants and mutations in S protein receptor-binding domain (RBD) of Omicron BA.2.12.1, BA.4 and BA.5 were described. The binding affinity of Omicron S protein RBD to heparin was further investigated by surface plasmon resonance (SPR). Solution competition studies on the inhibitory activity of heparin oligosaccharides and desulfated heparins at different sites on S protein RBD–heparin interactions revealed that different sub-lineages tend to bind heparin with different chain lengths and sulfation patterns. Furthermore, blind docking experiments showed the contribution of basic amino acid residues in RBD and sulfo groups and carboxyl groups on heparin to the interaction. Finally, pentosan polysulfate and mucopolysaccharide polysulfate were evaluated for inhibition on the interaction of heparin and S protein RBD of Omicron BA.2.12.1, BA.4/BA.5, and both showed much stronger inhibition than heparin.

## 1. Introduction

Since the beginning of the COVID-19 pandemic, numerous mutations of severe acute respiratory syndrome coronavirus 2 (SARS-CoV-2) have been identified and shared on GISAID (Global Initiative on Sharing Avian Influenza Data). A variant is recognized as a Variant of concern (VOC) by the World Health Organization (WHO) if it demonstrates: (i) increased transmissibility; (ii) detrimental change; (iii) increased in virulence; (iv) change in clinical disease presentation; (v) decreased effectiveness of public health and social measures of available diagnostics, vaccines and therapeutics. Previously circulating VOCs include Alpha, Beta, Gamma and Delta, while Omicron is currently the dominant variant circulating globally with greatly increased transmissibility [1]. Emergence of the Omicron variant has raised serious concerns about the increased risk of infectivity, immune evasion and reinfection. 

The Omicron variants include BA.1, BA.2, BA.3, BA.4, BA.5 and descendent lineages, but also BA.1/BA.2 circulating recombinant forms such as XE [1]. The genome of SARS-CoV-2 (~30 kb) encodes 16 non-structural proteins (NSPs) and 4 main structural proteins, including spike (S), envelope (E), core membrane (M), and nucleocapsid (N), and other accessory proteins [2]. Genome sequenced data of the Omicron variant demonstrated that the Omicron variant was the most highly mutated strain compared with the other VOCs, with 50 mutations accumulated throughout the genome and 26–32 mutations in the S protein [3]. Analysis of the mutations data shows that Omicron also carries several mutations found before, which were associated with increased infectivity and the chance of transmission by evading the immune response [4]. The current used COVID-19 vaccines mainly target the S protein [5]. Although several vaccines offering protection for COVID-19, studies showed a marked reduction the neutralizing capacity of vaccine induced immunity against the Omicron variant, especially the sub-lineages BA.4, and BA.5 [4]. Other therapeutics, including various monoclonal antibodies [6], remdesivir [7,8], tocilizumab [9], favipiravir [10], nirmatrelvir plus ritonavir (Paxlovid™) [11], molnupiravir and other approved drugs [12] have been used with different treatments [13]. With the virus mutation occurring so rapidly, alternative, or complementary approaches, need to be considered that require durable therapeutic effects and reduced adverse events and facilitate rapid development and large-scale production. 

Glycosaminoglycans (GAGs) are a class of linear polysaccharides, including heparin/ heparan sulfate (HS), keratan sulfate (KS), chondroitin sulfate (CS)/dermatan sulfate (DS), and hyaluronan (HA), and commonly expressed in the interior, cell surface, and extracellular environment of many cell types [14]. Pathogens exploit fundamental biological activities of GAGs, such as serving as cell adhesion and internalization receptors, inducing conformational changes, activating signaling pathways, to promote their attachment and invasion of host cells and to protect themselves from immune attack [15,16]. These activities suggest that GAGs are potential targets for the development of specific and effective antipathogen therapies. Studies have confirmed that SARS-CoV-2 interacts with both cellular HS and angiotensin-converting enzyme 2 (ACE2) through its receptor-binding domain (RBD) in the S1 subunit of the S protein. Binding of HS to S protein shifts the structure to favor the RBD open conformation that binds ACE2 [17,18]. Cellular HS acts as a co-factor for SARS-CoV-2 infection, this emphasizes the new therapeutic opportunities for targeting S protein–HS interactions. Studies have also shown that heparin may inhibit the activity of SARS-CoV-2 M^pro^ protein, thereby inhibiting virus replication and transcription, and heparin also reduces the activity of excessive heparanase, thereby inhibiting glycocalyx shedding and redox balance disturbance [19]. 

In previous work, we and others have shown that sulfated glycans, including heparin/HS, heparin derivatives, fucoidans, fucosylated chondroitin sulfate, and rhamnan sulfate inhibit the interaction between HS and the S protein RBD of wild type (WT), Delta variant and Omicron (B.1.1.529) [17,18,20,21,22,23,24]. Mutations occur in the RBD region of S protein may influence the binding to ACE2 or HS, for example, nine of the 15 RBD mutations in the Omicron (BA.1.1.529) Spike region belong to the binding footprint of the virus’ primary entry receptor [25]. Therefore, comparation of mutations in the emerging Omicron sub-lineages and the binding between their RBD and HS requires further analysis. It cannot be ignored that despite the positive effect of heparin on reducing the risk of venous thromboembolism and coagulopathy in COVID-19 patients, the risk of bleeding is increased [19]. In clinical use, heparin also has other side effect, such as heparin-induced thrombocytopenia [26]. Therefore, discovering the structural characteristic of heparin binding to the S-protein is critical for the development of therapeutics targeting S-protein– heparin interaction while reducing adverse effects. In this work, we examined the binding of the S protein RBD in Omicron sub-lineages BA.2.12.1, BA.4 and BA.5 with heparin, heparin oligosaccharides of different lengths, and chemically modified heparins using surface plasmon resonance (SPR) to elucidate the importance of size and sulfo group position for heparin/HS binding. We also performed blind docking experiments to objectively identify the preferred binding residues of heparin/HS and the associated amino acids on RBD region. Finally, highly negative compounds, including pentosan polysulfate (PPS) and mucopolysaccharide polysulfate (MPS) were evaluated for their inhibition of the RBD–heparin interaction.

## 2. Materials and Methods

### 2.1. Materials

S protein RBD of Omicron sub-lineages BA.2.12.1 (Cat: 40592-V08H132), BA.4/BA.5 (Cat: 40592-V08H130) were purchased from Sino Biological Inc. (Beijing, China).The proteins were constructed as follows: (1) a DNA sequence encoding the SARS-CoV-2 (BA.2.12.1) Spike RBD (YP_009724390.1, with mutations G339D, S371F, S373P, S375F, T376A, D405N, R408S, K417N, N440K, L452Q, S477N, T478K, E484A, Q493R, Q498R, N501Y, Y505H) (Arg319–Phe541) was expressed with a polyhistidine tag at the C-terminus; and (2) a DNA sequence encoding the SARS-CoV-2 (BA.4/BA.5) Spike RBD (YP_009724390.1, with mutations G339D, S371F, S373P, S375F, T376A, D405N, R408S, K417N, N440K, L452R, S477N, T478K, E484A, F486V, Q498R, N501Y, Y505H) (Arg319–Phe541) was expressed with a polyhistidine tag at the C-terminus. Unfractionated heparin (15 kDa) was purchased from Celsus Laboratories (Cincinnati, OH, USA). Desulfated heparins including *N*-desulfated heparin (14 kDa), 2-*O*-desulfated IdoA heparin (13 kDa), 6-*O*-desulfated heparin (13 kDa) and heparin oligosaccharides from tetrasaccharide (dp4) to octadecasaccharide (dp18) were purchased from Iduron (Manchester, UK). Pentosan polysulfate (PPS; 6.5 kDa) was from Bene Pharma (Munich, Germany). Mucopolysaccharide polysulfate (MPS; 14.5 kDa) was from Luitpold Pharma (Munich, Germany). Sensor SA chips were from Cytiva (Uppsala, Sweden). SPR experiments were performed using a BIAcore 3000 or T200 SPR (Cytiva, Uppsala, Sweden) with Biaevaluation software (version 4.0.1 or 3.2, Cytiva, Uppsala, Sweden).

### 2.2. Preparation of Heparin Biochips

The preparation of biotinylated heparin was as follows: heparin (2 mg) and amine-PEG3-Biotin (2 mg, Thermo Scientific, Waltham, MA, USA) were dissolved in 200 µL H_2_O added with 10 mg NaCNBH_3_, and reacted at 70 °C for 24 h, followed by additional 10 mg NaCNBH_3_ and reacted for another 24 h. The desalted biotinylated heparin was immobilized onto streptavidin (SA) chips based on the manufacturer’s protocol, as previously described [27].

### 2.3. Binding Kinetics and Affinity Measurement

S protein RBD of BA.2.12.1 and BA.4/BA.5 were diluted into HBS-EP+ buffer at concentrations of 1000, 500, 250, 125, and 63 nM, respectively. Diluted protein samples were injected at a flow rate of 30 µL/min for 3 min at 25 °C, followed by dissociation with HBS-EP+ buffer for 3 min. The sensor surface was regenerated by 2 M NaCl (30 µL) after each binding measurement. 

### 2.4. Evaluation of the Inhibition Activity of Heparin Oligosaccharides and Chemically Modified Heparins on S Protein RBD–Heparin Interaction Using Solution Competition SPR

Competition studies between surface-immobilized heparin and heparin analogues (heparin oligosaccharides and desulfated heparins) in solution mixed with S protein RBD, were performed as previously described [24]. S protein RBD samples (250 nM) were pre-mixed individually with 1000 nM oligosaccharides (dp4–dp18) or desulfated heparins and injected at a flow rate of 30 µL/min for 3 min at 25 °C. After dissociation, the sensor was regenerated by 2 M NaCl (30 µL). A control experiment (only S protein RBD) was used to test the complete regeneration. 

### 2.5. Model Building and Molecular Docking

Molecular docking and modeling of the S protein RBD with heparin dodecasaccharide were performed using AutoDock Vina. S protein RBD of Omicron (BA.2.12.1) was derived from the PDB library under code 7XNS, and the structure of dodecasaccharide, IdoA2S-GlcNS6S-IdoA2S-GlcNS6S-IdoA2S-GlcNS6S-IdoA2S-GlcNS6S-IdoA2S-GlcNS6S-IdoA2S-GlcNS6S, was derived from the NMR structure (PDB:1HPN). The structure of BA.4/BA.5 S protein RBD was derived from the mutation of BA.2.12.1 (PDB:7XNS) and optimized by the CHARMm force field. All hydrogen atoms were added to S protein RBD and charged using Gasteiger. A box of size (100,77,122) and grid center (42.110,182.058,131.292) was built for ligand docking. The RBD was in the center of the box and covered completely. The dodecaccharide was allowed to move freely in the box. During blind docking, all monosaccharide rings can rotate freely, and ring substituents (such as the sulfonic acid group) are defined as flexible and free to rotate. After molecular docking simulations, the binding poses were assessed based on binding energy in kcal/mol, and the low energy binding pose (more stable conformer) was chosen.

### 2.6. Evaluation of the Inhibition Activity of PPS and MPS on S Protein RBD–Heparin Interaction Using Solution Competition SPR

Likewise, SARS-CoV-2 S protein RBD samples (250 nM) pre-mixed with 1000 nM PPS or MPS were injected at a flow rate of 30 µL/min. The signal (RU) decreased when the binding sites on the S protein RBD were occupied by PPS or MPS instead of the surface-immobilized heparin. 

## 3. Results and Discussion

### 3.1. SARS-CoV-2 Variants and Omicron S Protein RBD Mutations

On November 2021, the WHO defined B.1.1.529 as the fifth VOC and named it Omicron. Multiple new subvariants/sub-lineages of Omicron have now emerged causing a significant increase in COVID-19 cases. High-throughput sequencing technologies enabled rapid identification of SARS-CoV-2 variants. The overall relationships of SARS-CoV-2 variants over time and the VOCs are shown in Figure 1A. Orange and red nodes are Omicron and its sub-lineages, where the red nodes are the three sub-lineages studied in this work, BA.2.12.1, BA.4 and BA.5. Sequence comparison showed that mutations in Omicron were mostly restricted to the S and N proteins, while other viral proteins were generally conserved [28]. The sequence alignment of the S protein RBD (Arg319–Phe541) of Omicron sub-lineages is shown in Figure 1B. An * (asterisk) represents shared mutations, and all the three sub-lineages have 13 amino acid mutations in RBD compared to WT, although the mutation positions are not identical. Notably, the mutations in BA4 and BA5 RBD are the same. The positively charged mutations in BA.2.12.1 RBD are N440K, T478K, Q493R, Q498R and Y505H, while in BA.4/BA.5 RBD are N440K, L452R, T478K, Q498R and Y505H. Persistent amino acid mutations will make it more difficult to provide rapid and reliable diagnosis and treatment. 

### 3.2. Binding Affinity and Kinetics Measurement on S Protein RBD–Heparin Interactions

Heparin/HS has variable repeating units, L-iduronic acid (IdoA) or D-glucuronic acid (GlcA) linked to *N*-sulfoglucosamine (GlcNS) or *N*-acetylglucosamine (GlcNAc), with different sulfo group modification [30]. Heparin/HS and other GAGs interact with proteins mainly through their highly negatively charged groups (sulfo groups and carboxyl groups) in polysaccharide chains binding to basic amino acid residues of proteins, for which a limited number of specific binding cases have, thus far, been discovered [31]. HS interacts with SARS-CoV-2 S protein and facilitates host cell entry of SARS-CoV-2 as a co-receptor of ACE2 [17]. Destabilizing and stabilizing mutations may have a large impact on the structure and pathogenesis of the virus. Since the Omicron S protein RBD have a more positive electrostatic potential than both WT and Delta [32], the binding of Omicron BA.2.12.1 and BA.4/BA.5 to heparin/HS is further investigated in the current study. 

SPR was used to measure the kinetics and binding affinity of SARS-CoV-2 S protein RBD interaction with heparin, a highly sulfated version of HS. Sensorgrams of S-protein RBD (BA.2.12.1 and BA.4/BA.5) interactions with immobilized heparin are shown in Figure 2. The sensorgrams were used to determine kinetics and binding affinity (i.e., association rate constant, k_a_; dissociation rate constant, *k_d_*; and binding equilibrium dissociation constant, K_D_, where K_D_ = *k_d_/k_a_*) by globally fitting the entire association and dissociation phases using a 1:1 Langmuir-binding model (Table 1). The binding affinities of S protein RBDs (BA.2.12.1 and BA.4/BA.5) were all nanomolar, which were slightly stronger than that of WT measured in our previous work (K_D_ = 400 nM) and comparable to that of Delta (K_D_ = 140 nM) and Omicron B.1.1.529 (K_D_ = 100 nM) [24]. Interestingly, Omicron BA.4/BA.4 RBD had a lower affinity (K_D_ = 230 nM) for heparin than Omicron BA.2.12.1 RBD (K_D_ = 140 nM), although they carry the same number of basic amino acids, albeit in different sequences.

### 3.3. Solution Competition Study on the Inhibition Activity of Heparin Oligosaccharides and Chemically Modified Heparins on S Protein RBD–Heparin Interaction

The molecular and biophysical properties of S protein–GAG binding is currently being investigated. For previous SARS-CoV-2 variants, either the sulfation pattern or chain length of HS/heparin showed an effect on binding affinity. HS with a higher degree of sulfation showed a higher affinity toward SARS-CoV-2 S protein subunits, a full-length molecule and its trimer, and the binding was also positively related to the 6-*O*-sulfation level [33]. Kim et al. showed the level of sulfation had critical impact on the SARS-CoV-2–GAG interaction, and the binding preferred long, highly sulfated structures [18]. For Omicron (BA.1.1.529), heparin showed size-dependent inhibition on the binding to S protein RBD, while higher sulfation level in heparin may not be that important for binding Omicron S protein RBD [24].

Solution competition was applied to test the effect of the chain length and sulfation pattern of heparin on the heparin interactions with RBD of Omicron BA.2.12.1, BA.4/BA.5 S proteins. S protein RBD was pre-mixed with different concentrations of heparin oligosaccharides or chemically desulfated heparins, and then injected onto the heparin chip. The signal (RU) decreased when the binding sites on the S protein RBD were occupied by PPS or MPS instead of the surface-immobilized heparin. Different heparin oligosaccharides at 1000 nM were applied in the competition analysis (Figure 3A,B and Figure 4A,B). For Omicron BA.2.12.1 S protein RBD, heparin oligosaccharides (from dp4 to dp18) showed a weak (2–27% reduction) and size-dependent inhibition on the binding. In the case of Omicron BA.4/BA.5, heparin oligosaccharides in solution competed more effectively (9–35% reduction) against S protein RBD binding to the heparin chip and was independent of chain length. The ability of different chemically desulfated heparins to inhibit the interaction of S protein with surface-immobilized heparin was also measured. All three chemically desulfated heparins reduced the binding of all three S-protein RBDs to surface-immobilized heparin (Figure 3C,D and Figure 4C,D). To our surprise, the binding biases of BA.2.12.1 and BA.4/BA.5 for sulfation patterns were quite different, with BA.4/BA.5 showed stronger charge-dependent and preference for 6-*O*-sulfation groups, possibly due to three amino acids differ in the RBD.

### 3.4. Molecular Modeling of the SARS-CoV-2 Spike RBD Interaction with Heparin

AutoDock Vina was used to construct a theoretical binding model of the Omicron S protein RBD and heparin oligosaccharides. Omicron BA.4 variant S protein, derived from the PDB library (PDB:7XNS), was shown in Figure 5A, with RBD domain in red. The RBD in Omicron BA.2.12.1 S protein was derived from the structure of Omicron BA.4/BA.5 RBD (PDB: 7XNS) and the dodecasaccharide, IdoA2S-GlcNS6S-IdoA2S-GlcNS6S-IdoA2S-GlcNS6S-IdoA2S-GlcNS6S-IdoA2S-GlcNS6S-IdoA2S-GlcNS6S, was derived from the NMR structure (PDB: 1HPN). The ranking of the binding poses was based on affinity energy, and the conformation with the lowest energy was selected for subsequent analysis of the S-protein RBD–heparin interaction. We report the best binding pose for BA.2.12.1 based upon lowest binding energy value (affinity energy = −4.2 kcal/mol, RMSD u.b. = 10.879), and the best binding pose for BA.4/BA5 was also selected by the same principle (affinity energy = −5.8 kcal/mol, RMSD u. b. = 31.459). The optimal binding model of the dodecasaccharide binding to BA.2.12.1 or BA.4/BA.5 S protein RBD was displayed using Pymol, and the electrostatic potential map of the binding conformation is shown in Figure 5B. The binding sites of heparin dodecasaccharide to BA.2.12.1 and BA.4/BA.5 were both located at or near the basic amino acid-rich domain, although different amino acid sites were chosen for binding. As shown in Figure 5C, the interaction of dodecasaccharide with BA.2.12.1 is dominated by electrostatic forces, assisted by hydrogen bonds, and both 6-*O*-sulfation and carboxyl groups on heparin chain contribute to the binding. The interaction of the dodecasaccharide and BA.4/BA.5 is governed by a combination of electrostatic forces and hydrogen bonding. The sulfate groups and carboxyl groups on heparin strengthen the interaction. The amino acid residues, such as R355, R577 and R357 in BA.2.12.1 RBD and R346, K440, K444 in BA.4/BA.5 RBD, make up a potential binding site for heparin and heparan sulfate, and other amino acids residues shown in Figure 5C also further strengthen the interaction.

### 3.5. Potential Anti-SARS-CoV-2 Activity of Pentosan Polysulfate and Mucopolysaccharide Polysulfate

Previously work showed that the structural heparin analogues, PPS and MPS, showed strong binding affinities to S protein in isothermal fluorescence titration and surface plasmon resonance (SPR) experiments [27,34,35]. PPS exhibits reduced anticoagulant potential [35] and is less likely to induce bleeding complications for long-term and high-dose use. Based on these, we investigated the ability of PPS and MPS to inhibit Omicron S protein RBD–heparin binding. The structure of PPS and MPS are shown in Figure 6A, the high level of sulfo groups enable strong interaction with S protein RBD. Both PPS and MPS in solution showed remarkable inhibition activity against surface-immobilized heparin binding with the Omicron S-protein RBD, stronger than that of heparin in solution (positive control). PPS and MPS potently inhibited the S protein RBD (BA.2.12.1)–heparin interaction by 99% and 89% (Figure 6B,C), respectively, while inhibiting the S protein RBD (BA.4/BA.5)–heparin interaction by 92% and 80% (Figure 6D,E), respectively. PPS and MPS showed promise for potential therapeutic or preventive agents against COVID-19. 

Solution competition dose response analysis was performed to calculate IC_50_ values to examine the ability of PPS and MPS to inhibit the interaction between surface-immobilized heparin with the S-protein RBD of Omicron variants. Once the active binding site on the S-protein RBD is occupied by glycan in solution, its binding to the surface-immobilized heparin is reduced, resulting in a concentration-dependent decrease in signal. IC_50_ measurement of the inhibition of S-protein RBD of Omicron BA.2.12.1 binding to surface-immobilized heparin using solution competition SPR by PPS and MPS were shown in Figure 7. Measured IC_50_ (concentration of competing analyte resulting in a 50% decrease in RU) = 228.9 nM, 19.3 nM, and 9.7 nM for heparin, PPS, and MPS, respectively. The IC_50_ values of the inhibition of S-protein RBD of Omicron BA.4/BA.5 binding to surface-immobilized heparin were also measured using solution competition SPR by PPS and MPS (Figure 8). The IC_50_ values of heparin, PPS and MPS were 680.1 nM, 84.4 nM, and 124.9 nM, respectively. Although both BA2.12.1 and BA.4/BA.5 were strongly inhibited by PPS and MPS, the activity of BA.4/BA.5 appeared to be more difficult to be inhibited by sulfated glycans.

## 4. Conclusions

Mutations occurring in the RBD region of SARS-CoV-2 S protein may influence the binding to HS/heparin. SPR analysis revealed that the binding affinity (K_D_) of Omicron BA.2.12.1 and BA.4/BA.5 were all at nanomolar concentrations, which were slightly stronger than that of WT and comparable to that of Delta and Omicron B.1.1.529. Solution competition studies indicated that efficient binding of Omicron BA.2.12.1 requires longer chain length, which is not that necessary for BA.4/BA.5. Competition assays also demonstrated that all the sulfation sites are important for interaction between the S protein RBDs and heparin, although higher sulfation level of HS/heparin are required for binding to BA.4/BA.5. The three sub-lineages showed differences in binding models for heparin dodecasaccharide, suggesting that mutations in RBD have an important effect on viral attachment, possibly explaining differences in the SARS-CoV-2 infection. Binding of human ACE2 to S protein RBD from Omicron and Delta was studied by Han and co-workers, showing that Omicron, Delta, and WT SARS-CoV-2 RBDs have similar binding strengths to hACE2 [32]. Therefore, it is also promising to investigate the effect of different glycans on the binding of ACE2 to different RBDs. Although, persistent amino acid mutations will make it more difficult to provide rapid and reliable diagnosis and treatment, PPS and MPS show promise as therapeutic and/or preventative antiviral drugs against COVID-19.

## Figures and Tables

**Figure 1 viruses-14-02696-f001:**
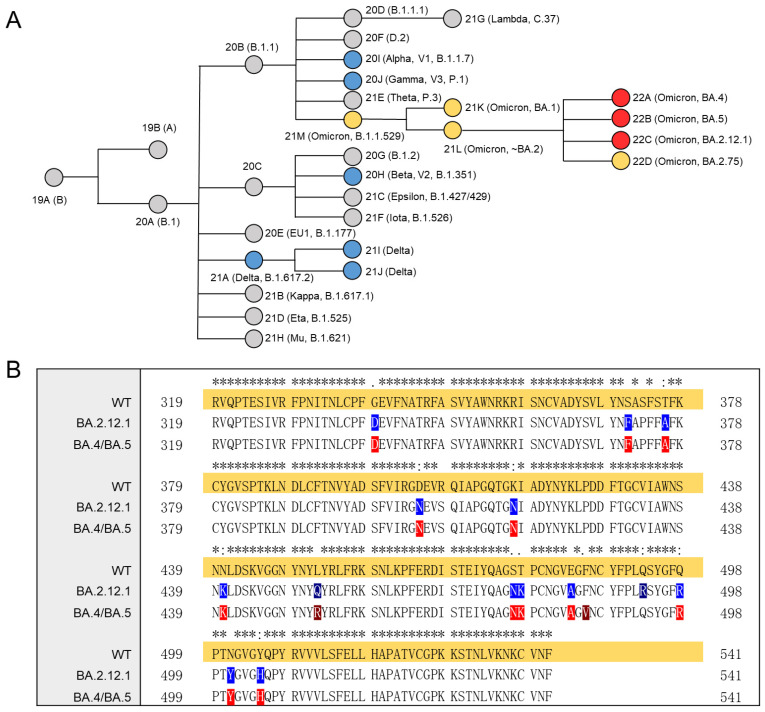
Phylogenetic tree and multiple sequence alignment. (**A**) Phylogenetic relationships of Nextstrain SARS-CoV-2 clades. The phylogenetic tree was adapted from figure provided by Nextstrain and CoVariants [29]. VOCs are represented by colored nodes. (**B**) Mutation profile of S protein RBD of Omicron BA.2.12.1, BA.4/BA.5 compared with WT. Multiple sequence alignment was performed by Clustal Omega (1.2.4). An * (asterisk) indicates positions which have a single, fully conserved residue.

**Figure 2 viruses-14-02696-f002:**
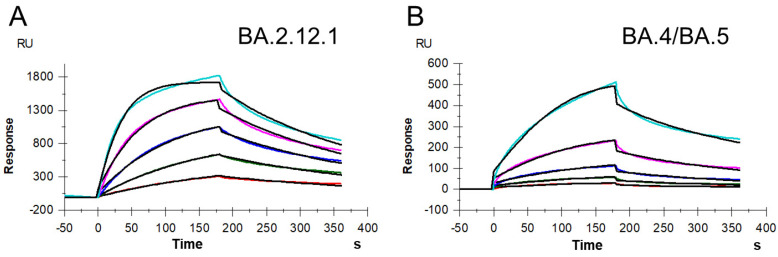
SPR sensorgrams of S protein RBD of BA.2.12.1 and BA.4/BA.5 binding with heparin. (**A**) SPR sensorgrams of S protein RBD of BA.2.12.1 binding with heparin. Concentrations of RBD (from top to bottom) are 1000, 500, 250, 125, and 63 nM, respectively. (**B**) SPR sensorgrams of S protein RBD of BA.4/BA.5 binding with heparin. Concentrations of RBD (from top to bottom) are 1000, 500, 250, 125, and 63 nM, respectively.

**Figure 3 viruses-14-02696-f003:**
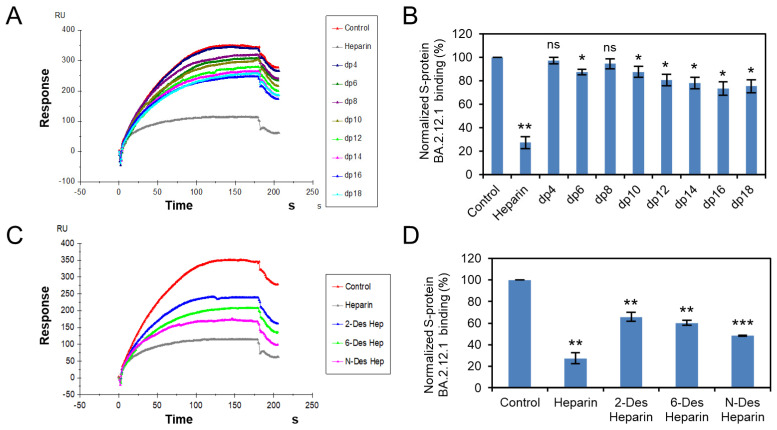
S protein RBD (BA.2.12.1)–heparin interaction inhibited by heparin oligosaccharides/desulfated heparins using solution competition. (**A**) SPR sensorgrams of S protein RBD (BA.2.12.1)–heparin interaction competing with different heparin oligosaccharides. Concentration of S-protein RBD (BA.2.12.1) is 250 nM mixed with 1 µM of different heparin oligosaccharides. (**B**) Bar graphs (based on triplicate experiments with standard deviation) of normalized S-protein RBD (BA.2.12.1) binding preference to surface heparin by competing with different heparin oligosaccharides. (**C**) SPR sensorgrams of S protein RBD (BA.2.12.1)–heparin interaction competing with different desulfated heparins. Concentration of S-protein RBD (BA.2.12.1) is 250 nM mixed with 1 µM of different desulfated heparins. (**D**) Bar graphs (based on triplicate experiments with standard deviation) of normalized S-protein RBD (BA.2.12.1) binding preference to surface heparin by competing with different desulfated heparins. Statistical analysis was performed using unpaired two-tailed *t*-test (ns: *p* > 0.05 compared to the control, *: *p* ≤ 0.05 compared to the control, **: *p* ≤ 0.01 compared to the control, ***: *p* ≤ 0.001 compared to the control).

**Figure 4 viruses-14-02696-f004:**
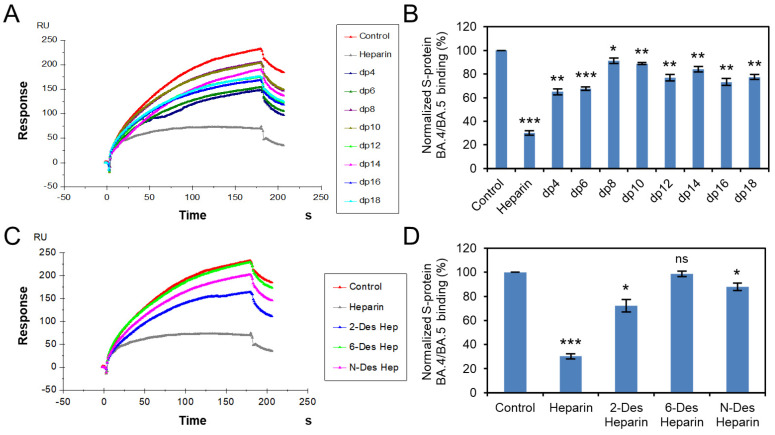
S protein RBD (BA.4/BA.5)–heparin interaction inhibited by heparin oligosaccharides/desulfated heparins using solution competition. (**A**) SPR sensorgrams of S protein RBD (BA.4/BA.5)–heparin interaction competing with different heparin oligosaccharides. Concentration of S-protein RBD (BA.4/BA.5) is 250 nM mixed with 1 µM of different heparin oligosaccharides. (**B**) Bar graphs (based on triplicate experiments with standard deviation) of normalized S-protein RBD (BA.4/BA.5) binding preference to surface heparin by competing with different heparin oligosaccharides. (**C**) SPR sensorgrams of S protein RBD (BA.4/BA.5)–heparin interaction competing with different desulfated heparins. Concentration of S-protein RBD (BA.4/BA.5) is 250 nM mixed with 1 µM of different desulfated heparins. (**D**) Bar graphs (based on triplicate experiments with standard deviation) of normalized S-protein RBD (BA.4/BA.5) binding preference to surface heparin by competing with different desulfated heparins. Statistical analysis was performed using unpaired two-tailed *t*-test (ns: *p* > 0.05 compared to the control, *: *p* ≤ 0.05 compared to the control, **: *p* ≤ 0.01 compared to the control, ***: *p* ≤ 0.001 compared to the control).

**Figure 5 viruses-14-02696-f005:**
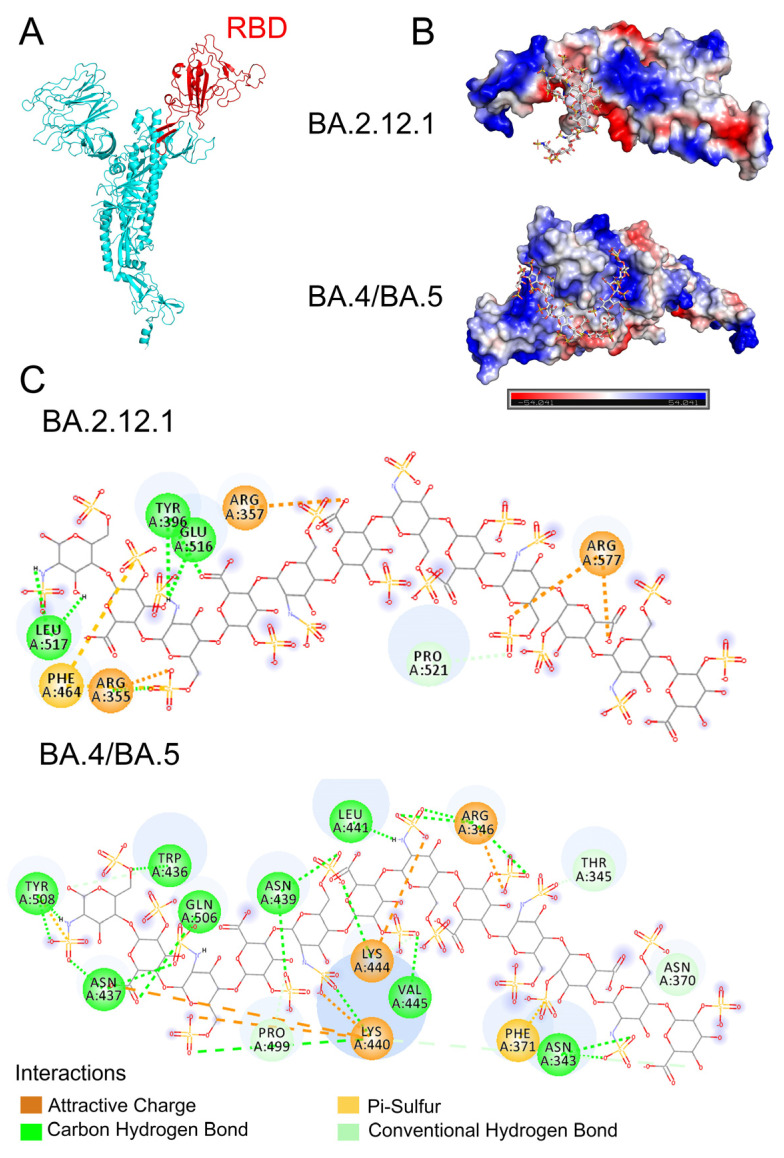
Molecular Docking and modeling simulation. (**A**) Structure of Omicron S protein (PDB: 7XNS) with the RBD domain in red. (**B**) Model electrostatic potential map for docking binding of BA.2.12.1 and BA.4/BA.5 S protein RBD to heparin dodecasaccharide (PDB:1HPN). (**C**) 2D diagram of the interaction of BA.2.12.1 and BA.4/BA.5 S protein RBDs with heparin dodecasaccharide.

**Figure 6 viruses-14-02696-f006:**
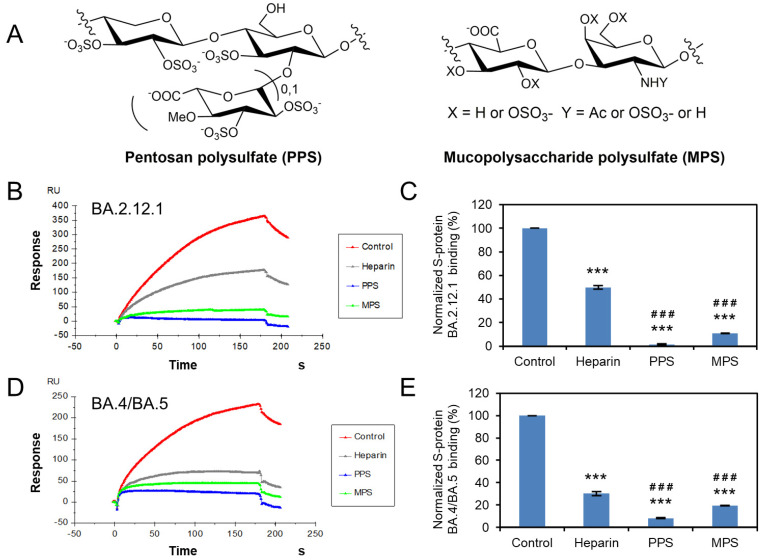
Solution competition between heparin and PPS or MPS. (**A**) Structure of PPS and MPS. (**B**) SPR sensorgrams of S protein RBD (BA.2.12.1)–heparin interaction competing with PPS or MPS. Concentration of S-protein RBD (BA.2.12.1) is 250 nM mixed with 1 µM of PPS or MPS. (**C**) Bar graphs (based on triplicate experiments with standard deviation) of normalized S-protein RBD (BA.2.12.1) binding preference to surface heparin by competing with PPS or MPS. (**D**) SPR sensorgrams of S protein RBD (BA.4/BA.5)–heparin interaction competing with PPS or MPS. Concentration of S-protein RBD (BA.4/BA.5) is 250 nM mixed with 1 µM of PPS or MPS. (**E**) Bar graphs (based on triplicate experiments with standard deviation) of normalized S-protein RBD (BA.4/BA.5) binding preference to surface heparin by competing with PPS or MPS. Statistical analysis was performed using unpaired two-tailed *t*-test (***: *p* ≤ 0.001 compared to the control, ###: *p* < 0.001 compared to the heparin).

**Figure 7 viruses-14-02696-f007:**
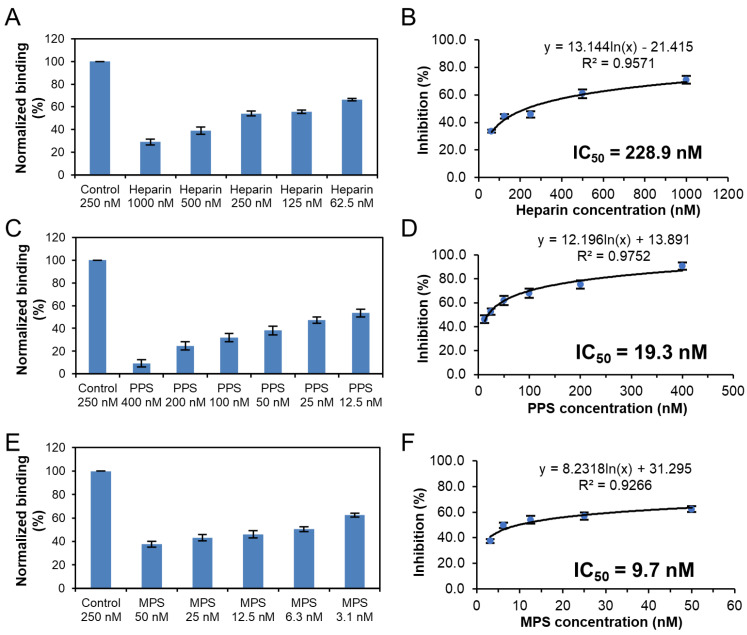
IC_50_ measurement of the inhibition of S-protein RBD (BA.2.12.1) binding to heparin using solution competition SPR by sulfated glycans (heparin, PPS, and MPS). S-protein RBD concentration was 250 nM. Error bars represent standard deviations from triplicate tests. (**A**,**B**) = heparin; (**C**,**D**) = PPS; (**E**,**F**) = MPS.

**Figure 8 viruses-14-02696-f008:**
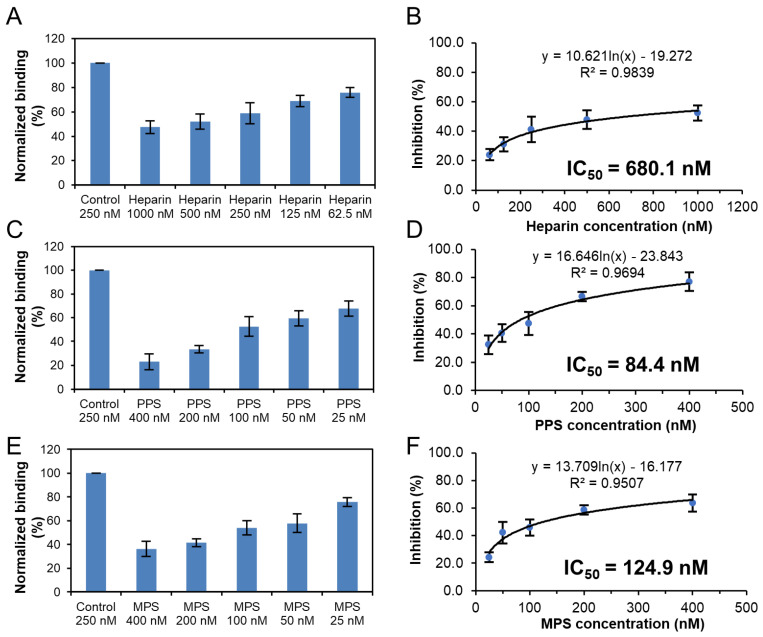
IC_50_ measurement of the inhibition of S-protein RBD (BA.4/BA.5) binding to heparin using solution competition SPR by sulfated glycans (heparin, PPS, and MPS). S-protein RBD concentration was 250 nM. Error bars represent standard deviations from triplicate tests. (**A**,**B**) = heparin; (**C**,**D**) = PPS; (**E**,**F**) = MPS.

**Table 1 viruses-14-02696-t001:** Summary of kinetic data of S protein RBD of BA.2.12.1 and BA.4/BA.5 binding with heparin.

	*k_a_* (M^−1^s^−1^)	*k_d_* (s^−1^)	*K_D_* (M)
BA.2.12.1	3.4 × 10^4^ ( ± 270) *	4.7 × 10^−3^ ( ± 2.1 × 10^−5^) *	1.4 × 10^−7^ ( ± 5.8 × 10^−9^) **
BA.4/BA.5	3.4 × 10^4^ ( ± 630)	7.9 × 10^−3^ ( ± 8.7 × 10^−5^)	2.3 × 10^−7^ ( ± 2.6 × 10^−8^)

* The data with (±) in parentheses are the standard deviations (SD) from global fitting of five injections. ** Standard deviation (SD) on triplicated experiments.

## Data Availability

Not applicable.

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
