# Peer review of "Structural Characteristics of Heparin Binding to SARS-CoV-2 Spike Protein RBD of Omicron Sub-Lineages BA.2.12.1, BA.4 and BA.5"

_viruses, 2022, doi:10.3390/v14122696_

Round 1

Reviewer 1 Report

The authors presented the structural basis for heparin binding to SARS-CoV-2 spike pro- 2 tein RBD of Omicron sub-lineages BA.2.12.1, BA.4 and BA.5. I recommend publication after addressing the following points:

The authors need to discuss the role of the inhibition of heparanase, an endothelial glycocalyx-degrading enzyme that contributes to vascular leakage and inflammation. Please, find Mangiafico, Marco, Andrea Caff, and Luca Costanzo. "The Role of Heparin in COVID-19: An Update after Two Years of Pandemics." Journal of Clinical Medicine 11.11 (2022): 3099.

The authors need to briefly discuss the bleeding risk of heprin

The authors need to specify which heparin was studied, Unfractionated or Low-Molecular-Weight Heparin

Author Response

The authors presented the structural basis for heparin binding to SARS-CoV-2 spike protein RBD of Omicron sub-lineages BA.2.12.1, BA.4 and BA.5. I recommend publication after addressing the following points:

The authors need to discuss the role of the inhibition of heparanase, an endothelial glycocalyx-degrading enzyme that contributes to vascular leakage and inflammation. Please, find Mangiafico, Marco, Andrea Caff, and Luca Costanzo. "The Role of Heparin in COVID-19: An Update after Two Years of Pandemics." Journal of Clinical Medicine 11.11 (2022): 3099.

Response: We thank the reviewer for the positive and constructive comments. We added this reference and introduce this work in the introduction section, Line 75.

The authors need to briefly discuss the bleeding risk of heparin.

Response: We added a brief description of the increased bleeding risk in COVID-19 patients treated with heparin and other adverse effect of heparin in clinical use to emphasize the importance of S-protein RBD–heparin interaction study, Line 87.

The authors need to specify which heparin was studied, Unfractionated or Low-Molecular-Weight Heparin

Response: The heparin used in this study was unfractionated heparin with a molecular weight of 15 kDa. We added “Unfractionated” to specify with the heparin, Line 112.

Reviewer 2 Report

This manuscript reports the three spikes of Omicron sub-lineages (BA.2.12.1, BA.4, BA.5) show different binding modes for heparin dodecasaccharide, mainly using SPR-based competitive assays and molecular docking analysis, which emphasized the importance of the chain length and sulfation of heparin for establishing interactions with the SARS-CoV-2 spike proteins. In this study, it is found that BA.2.12.1 spike preferentially binds to the heparin with longer chain length, whereas higher sulfation level of HS is critical for its binding to BA.4/5 spike. Overall, the work may provide some better understanding of the increasing infection ability of Omicron subvariants and it appears to show some promise of new target for combating SARS-CoV-2. However, the experimental evidence is somewhat limited. Thus, there are some major points that should be addressed before the work can be consider for publication.

1. Considering the topic of your study, the tile of the paper which focused on the “structure basis”, is not well-suited to the key findings of main context. In addition, the so-called structure basis revealed by the structural analysis and molecular docking is insufficient to highlight and cover the main content of this study. So I would suggest the authors modify the tile.

2. Although this study focused on the Omicron subvariants BA.2.12.1 and BA.4/5, as for the affinity assays, the binding of the prototype RBD to the heparin should be included as an important control, which may contribute to provide an enough comparison due to the influence in the accumulated mutations. Also, it is not appropriate to formally compare the present RBD-heparin binding affinities results with those published by others before (Line 194). Additional SPR experiment for WT and Delta RBDs is necessary.

3. Figure 3, 4 and 6: As for the bar graphs, proper statistical analysis should be conducted to validate the difference as claimed.

4. All the activity experiments in this study were dependent on in vitro identification. A series of heparin oligosaccharides and chemically modified heparins should be validated, at least, for their inhibitive abilities against SARS-CoV-2 peudotyped viruses at the cellular level. 

5. The previous studies have shown that HS/heparin served as co-receptor for SARS-Cov-2 infection, and heparin binding to SARS-CoV-2 spike protein could increase its binding to ACE2. I wonder if different types of heparin may result in different affect effect on the binding of ACE2 to different RBDs. If so, it may explain why Omicron subvariants tend to bind certain types of heparin. Thus, some support experiment regarding the interaction relationship among S protein, heparin and ACE2 should be considered.

6. Figure 5C: The labels of residues in the Fig. 2D are far too small to be legible. Please make it clearer.

7.  Line 141: Detailed description about how authors filter the Vina docking results (scoring or clustering?) is needed.

Author Response

This manuscript reports the three spikes of Omicron sub-lineages (BA.2.12.1, BA.4, BA.5) show different binding modes for heparin dodecasaccharide, mainly using SPR-based competitive assays and molecular docking analysis, which emphasized the importance of the chain length and sulfation of heparin for establishing interactions with the SARS-CoV-2 spike proteins. In this study, it is found that BA.2.12.1 spike preferentially binds to the heparin with longer chain length, whereas higher sulfation level of HS is critical for its binding to BA.4/5 spike. Overall, the work may provide some better understanding of the increasing infection ability of Omicron subvariants and it appears to show some promise of new target for combating SARS-CoV-2. However, the experimental evidence is somewhat limited. Thus, there are some major points that should be addressed before the work can be consider for publication.

  1. Considering the topic of your study, the tile of the paper which focused on the “structure basis”, is not well-suited to the key findings of main context. In addition, the so-called structure basis revealed by the structural analysis and molecular docking is insufficient to highlight and cover the main content of this study. So I would suggest the authors modify the tile.

Response: Thanks for your positive comments and great suggestion. We have replaced “structural basis” with “structural characteristics”.

  1. Although this study focused on the Omicron subvariants BA.2.12.1 and BA.4/5, as for the affinity assays, the binding of the prototype RBD to the heparin should be included as an important control, which may contribute to provide an enough comparison due to the influence in the accumulated mutations. Also, it is not appropriate to formally compare the present RBD-heparin binding affinities results with those published by others before (Line 194). Additional SPR experiment for WT and Delta RBDs is necessary.

Response: The SPR analysis in the interaction with WT, Delta and Omicron B.1.1.529 RBDs mentioned in this manuscript were all previously published from our group. The results are repeatable. We add a description of “our previous work” in Line 215.

Reference [24]: Gelbach, A. L.; Zhang, F.; Kwon, S. J.; Bates, J. T.; Farmer, A. P.; Dordick, J. S.; Wang, C.; Linhardt, R. J., Interactions between heparin and SARS-CoV-2 spike glycoprotein RBD from omicron and other variants. Front Mol Biosci 2022, 9, 912887.

  1. Figure 3, 4 and 6: As for the bar graphs, proper statistical analysis should be conducted to validate the difference as claimed.

Response: We added the statistical analysis of Figure 3, 4 and 6 and validated the difference. The inhibition results of different glycans were compared with controls. Statistical analysis was performed using unpaired two-tailed t-test (ns: P > 0.05 compared to the control, *: P ≤ 0.05 compared to the control, **: P ≤ 0.01 compared to the control, ***: P ≤ 0.001 compared to the control).

  1. All the activity experiments in this study were dependent on in vitro identification. A series of heparin oligosaccharides and chemically modified heparins should be validated, at least, for their inhibitive abilities against SARS-CoV-2 peudotyped viruses at the cellular level. 

Response: We and other researchers did activity experiments for heparin/HS or other glycans inhibitive abilities against SARS-CoV-2 pseudo type viruses at the cellular level in previously work, shown as below. In this study, we focused on the structural characteristics of RBD protein rather than the whole viruses binding to heparin.

References:

[23]: Song, Y.; He, P.; Rodrigues, A. L.; Datta, P.; Tandon, R.; Bates, J. T.; Bierdeman, M. A.; Chen, C.; Dordick, J.; Zhang, F.; Linhardt, R. J., Anti-SARS-CoV-2 Activity of Rhamnan Sulfate from Monostroma nitidum. Mar Drugs 2021, 19, (12).

[27]: Zhang, F.; He, P.; Rodrigues, A. L.; Jeske, W.; Tandon, R.; Bates, J. T.; Bierdeman, M. A.; Fareed, J.; Dordick, J.; Linhardt, R. J., Potential Anti-SARS-CoV-2 Activity of Pentosan Polysulfate and Mucopolysaccharide Polysulfate. Pharmaceuticals (Basel) 2022, 15, (2).

[35]: Bertini, S.; Alekseeva, A.; Elli, S.; Pagani, I.; Zanzoni, S.; Eisele, G.; Krishnan, R.; Maag, K. P.; Reiter, C.; Lenhart, D.; Gruber, R.; Yates, E. A.; Vicenzi, E.; Naggi, A.; Bisio, A.; Guerrini, M., Pentosan Polysulfate Inhibits Attachment and Infection by SARS-CoV-2 In Vitro: Insights into Structural Requirements for Binding. Thromb Haemost 2022, 122, (6), 984-997.

  1. The previous studies have shown that HS/heparin served as co-receptor for SARS-Cov-2 infection, and heparin binding to SARS-CoV-2 spike protein could increase its binding to ACE2. I wonder if different types of heparin may result in different affect effect on the binding of ACE2 to different RBDs. If so, it may explain why Omicron subvariants tend to bind certain types of heparin. Thus, some support experiment regarding the interaction relationship among S protein, heparin and ACE2 should be considered.

Response: We thank the reviewer for the valuable suggestion. Binding of Human ACE2 to spike RBD from omicron and delta SARS-CoV-2 have been studied by Han and co-workers. The authors mentioned that Omicron, delta, and prototype SARS-CoV-2 RBDs showed similar binding strength to hACE2. We agree that it would be a very interesting project to investigate the effect of different types of heparins on the binding of ACE2 to different RBDs. We have added the discussion in the conclusion section, Line 374.

Reference:

[32] Han, P.; Li, L.; Liu, S.; Wang, Q.; Zhang, D.; Xu, Z.; Han, P.; Li, X.; Peng, Q.; Su, C.; Huang, B.; Li, D.; Zhang, R.; Tian, M.; Fu, L.; Gao, Y.; Zhao, X.; Liu, K.; Qi, J.; Gao, G. F.; Wang, P., Receptor binding and complex structures of human ACE2 to spike RBD from omicron and delta SARS-CoV-2. Cell 2022, 185, (4), 630-640 e10.

  1. Figure 5C: The labels of residues in the Fig. 2D are far too small to be legible. Please make it clearer.

Response: Thank you. We have revised the Figure based on your suggestion.

  1. Line 141: Detailed description about how authors filter the Vina docking results (scoring or clustering?) is needed.

Response: We added details to Docking Methods, “After molecular docking simulations, the binding poses were assessed based on binding energy in kcal/mol, and the low energy binding pose (more stable conformer) was chosen.”, Line 155.

    We also added description in the result section, “The ranking of the binding poses was based on affinity energy, and the conformation with the lowest energy was selected for subsequent analysis of the S-protein RBD–heparin interaction. We report the best binding pose for BA.2.12.1 based upon lowest binding energy value (affinity energy = –4.2 kcal/mol, RMSD u.b. = 10.879), and the best binding pose for BA.4/BA5 was also selected by the same principle (affinity energy = –5.8 kcal/mol, RMSD u. b. = 31.459).”, Line 293.

Reviewer 3 Report

Paper : Structural basis for heparin binding to SARS-CoV-2 spike protein RBD of Omicron sub-lineages BA.2.12.1, BA.4 and BA.5

Authors : Deling Shi  Changkai Bu  , Peng He  , Yuefan Song 2, Robert J. Linhardt  , Jonathan S. Dordick , Lianli Chi 1, and Fuming Zhang

General Comment

Based on previously published results, which reported on the binding of heparan sulfate to the RBD region of the spike (S) protein of SARS-CoV-2, in this paper Deling Shi and collaborators  have addressed the binding of heparin and heparin  derivatives  to the RBD of Omicron  subvariants of SARS-CoV-2, particularly the highly transmissible BA4/5 one. This has been done using several biochemical and biophysical approaches aimed at establishing nature and extent of the molecular interactions, as well as ways of inhibiting it within a perspective of generating new tools for SARS-CoV-2 control.

As it stands, the paper contains technically sound and scientifically useful information, although its originality is somehow tempered by the previous publications, and its applicability is greatly affected the lack of ex-vivo (cell-based) or in vivo (organism-or organoid-based) data helping the readers to have an idea of any potential impact of these results for SARS-CoV-2 fight. In particular, in solution studies with S protein -RBD or ACE2 inhibitors, though informative, may not be taken as exhaustive or even reliable indicators of inhibitory activity in cell-based or in vivo environment. This is a limitation that should be more clearly stated in the Discussion. 

Overall, while inviting the Authors to proceed with investigations aimed at filling the above gap in future studies,  I would also advise them to be extremely cautious on approaching in vivo studies of heparin and heparin derivatives or other similar biological products because of the critical physiological functions of these molecules.

Specific comments.

Please note that the absence of line numbers in the text makes always difficult offering precise notations or remarks by the reviewer (at least this reviewer!)

1.     Page 2, lines 9-10 above.  In mentioning antiviral therapeutics, it would be appropriate and informative to the readers including and quoting the recent Molnupiravir and Paxlovid anti-SARS-CoV2 approved and clinically used inhibitors

2.    Line before Materials and Methods; inhibition of, not on

3.    Page 6. The lines describing experiments with heparin immobilized report a weak reduction of binding by heparin oligosaccharides (2-27%) Indeed, in Figure 3 A-B, several inhibition bars appear to be minimally different, and no statistical assessment is being done. Generally, no statistics is used in this paper.  It appears the statistical variation is considered to be so little that no statistics is needed . I’m not sure about this: as an example, the biological plausibility of the inhibitory values of dp4 and dp8 as competitors of heparin binding is probably null. 

4.    More impressive are the inhibitory activities of PPS and MPS. However. a single inhibitor concertation (1 micromolar) is being used. Because of the importance of these data, a dose/effect curve could give a more informative output and easing comparison with previously published data with these inhibitors by the same Authors (Ref.23)

Author Response

Based on previously published results, which reported on the binding of heparan sulfate to the RBD region of the spike (S) protein of SARS-CoV-2, in this paper Deling Shi and collaborators have addressed the binding of heparin and heparin derivatives to the RBD of Omicron subvariants of SARS-CoV-2, particularly the highly transmissible BA4/5 one. This has been done using several biochemical and biophysical approaches aimed at establishing nature and extent of the molecular interactions, as well as ways of inhibiting it within a perspective of generating new tools for SARS-CoV-2 control.

As it stands, the paper contains technically sound and scientifically useful information, although its originality is somehow tempered by the previous publications, and its applicability is greatly affected the lack of ex-vivo (cell-based) or in vivo (organism-or organoid-based) data helping the readers to have an idea of any potential impact of these results for SARS-CoV-2 fight. In particular, in solution studies with S protein -RBD or ACE2 inhibitors, though informative, may not be taken as exhaustive or even reliable indicators of inhibitory activity in cell-based or in vivo environment. This is a limitation that should be more clearly stated in the Discussion.

Overall, while inviting the Authors to proceed with investigations aimed at filling the above gap in future studies, I would also advise them to be extremely cautious on approaching in vivo studies of heparin and heparin derivatives or other similar biological products because of the critical physiological functions of these molecules.

Specific comments.

Please note that the absence of line numbers in the text makes always difficult offering precise notations or remarks by the reviewer (at least this reviewer!)

  1. Page 2, lines 9-10 above. In mentioning antiviral therapeutics, it would be appropriate and informative to the readers including and quoting the recent Molnupiravir and Paxlovid anti-SARS-CoV2 approved and clinically used inhibitors

Response: We have added the recent inhibitors, Molnupiravir and Paxlovid, in the revised manuscript, Line 57.

  1. Line before Materials and Methods; inhibition of, not on

Response: We have revised this sentence, Line 99.

  1. Page 6. The lines describing experiments with heparin immobilized report a weak reduction of binding by heparin oligosaccharides (2-27%) Indeed, in Figure 3 A-B, several inhibition bars appear to be minimally different, and no statistical assessment is being done. Generally, no statistics is used in this paper. It appears the statistical variation is considered to be so little that no statistics is needed. I’m not sure about this: as an example, the biological plausibility of the inhibitory values of dp4 and dp8 as competitors of heparin binding is probably null.

Response: We added the statistical assessment of the inhibition data in the revised manuscript, Figure 3,4 and 6.

  1. More impressive are the inhibitory activities of PPS and MPS. However. a single inhibitor concertation (1 micromolar) is being used. Because of the importance of these data, a dose/effect curve could give a more informative output and easing comparison with previously published data with these inhibitors by the same Authors (Ref.23)

Response: We have added the IC50 data of heparin, PPS and MPS in Figure 7 and 8.

    We also added a discussion on these data in Line 340, “Solution competition dose response analysis was performed to calculate IC50 values to examine the ability of PPS and MPS to inhibit the interaction between surface-immobilized heparin with the S-protein RBD of Omicron variants. Once the active binding site on the S-protein RBD is occupied by glycan in solution, its binding to the surface-immobilized heparin is reduced, resulting in a concentration-dependent de-crease in signal. IC50 measurement of the inhibition of S-protein RBD of Omicron BA.2.12.1 binding to surface-immobilized heparin using solution competition SPR by PPS and MPS were shown in Figure 7. Measured IC50 (concentration of competing analyte resulting in a 50% decrease in RU) = 228.9 nM, 19.3 nM, and 9.7 nM for heparin, PPS, and MPS, respectively. The IC50 values of the inhibition of S-protein RBD of Omicron BA.4/BA.5 binding to surface-immobilized heparin were also measured using solution competition SPR by PPS and MPS (Figure 8). The IC50 values of heparin, PPS and MPS were 680.1 nM, 84.4 nM, and 124.9 nM, respectively. Although both BA2.12.1 and BA.4/BA.5 were strongly inhibited by PPS and MPS, the activity of BA.4/BA.5 seems be more difficult to be inhibited by sulfated glycans.”

Round 2

Reviewer 2 Report

The author have addressed all comments I raised.